

# Mitochondrial genomes of three Tetrigoidea species and phylogeny of Tetrigoidea

Li-Liang Lin[1], Xue-Juan Li[1], Hong-Li Zhang[2] and Zhe-Min Zheng[1]

[1] College of Life Sciences, Shaanxi Normal University, Xi'an, Shaanxi, China
[2] School of Life Sciences, Datong University, Datong, Shanxi, China

## ABSTRACT

The mitochondrial genomes (mitogenomes) of *Formosatettix qinlingensis*, *Coptotettix longjiangensis* and *Thoradonta obtusilobata* (Orthoptera: Caelifera: Tetrigoidea) were sequenced in this study, and almost the entire mitogenomes of these species were determined. The mitogenome sequences obtained for the three species were 15,180, 14,495 and 14,538 bp in length, respectively, and each sequence included 13 protein-coding genes (PCGs), partial sequences of rRNA genes (rRNAs), tRNA genes (tRNAs) and a A + T-rich region. The order and orientation of the gene arrangement pattern were identical to that of most Tetrigoidea species. Some conserved spacer sequences between trnS(UCN) and nad1 were useful to identify Tetrigoidea and Acridoidea. The Ka/Ks value of atp8 between *Trachytettix bufo* and other four Tetrigoidea species indicated that some varied sites in this gene might be related with the evolution of *T. bufo*. The three Tetrigoidea species were compared with other Caelifera. At the superfamily level, conserved sequences were observed in intergenic spacers, which can be used for superfamily level identification between Tetrigoidea and Acridoidea. Furthermore, a phylogenomic analysis was conducted based on the concatenated data sets from mitogenome sequences of 24 species of Orthoptera in the superorders Caelifera and Ensifera. Both maximum likelihood and bayesian inference analyses strongly supported Acridoidea and Tetrigoidea as forming monophyletic groups. The relationships among six Tetrigoidea species were (((((*Tetrix japonica*, *Alulatettix yunnanensis*), *Formosatettix qinlingensis*), *Coptotettix longjiangensis*), *Trachytettix bufo*), *Thoradonta obtusilobata*).

## INTRODUCTION

Tetrigoidea is a superfamily of Caelifera in Orthoptera and is regarded as a primitive taxon of Caelifera (*Cao & Zheng, 2011*). This superfamily contains approximately 274 genera and 2,356 species, according to the OSF website (Orthoptera Species File, http://orthoptera.speciesfile.org) (*Eades et al., 2014*). All species in Tetrigoidea are in the family Tetrigidae, which contains nine subfamilies (Batrachideinae, Cladonotinae, Cleostratinae, Discotettiginae, Lophotettiginae, Metrodorinae, Scelimeninae, Tetriginae and Tripetalocerinae) (*Eades et al., 2014*). Based on the morphological features of

Corresponding author
Xue-Juan Li, lixuejuan@snnu.edu.cn

antennae shape and the frontal ridge, anterior margin and lateral angle of the pronotum, the Tetrigoidea is divided into seven families by most Chinese taxonomists, i.e., Batrachididae, Cladonotidae, Discotettigidae, Metrodoridae, Scelimenidae, Tetrigidae and Tripetaloceridae (*Zheng, 2005*; *Deng, Zheng & Wei, 2007*).

Because of their small size and minor importance as agricultural pests, this group has been of little concern and the focus of few studies. The study of Tetrigoidea focused primarily on behaviour, morphology, anatomy and cytology before the 1990s, and included bioecological observations (*Paranjape & Bhalerao, 1985*) and karyology (*Warchałowska-Śliwa & Maryańska-Nadachowska, 1989*; *Maryańska-Nadachowska & Warchałowska-Śliwa, 1991*; *Del Cerro, Jones & Santos, 1997*; *Ma & Zheng, 1994*; *Ma & Guo, 1994*). Research on the molecular systematics of Tetrigoidea gradually appeared later, with most of the studies focusing only on single genes. For example, the phylogenetic results of *Flook & Rowell (1997a)* and *Flook & Rowell (1997b)* support the monophyly of Tetrigoidea and a close relation between Tetrigoidea and Tridactyloidea. In a study of the phylogeny of Tetrigoidea, *Jiang (2000)* showed that Scelimenidae was sister group to all other Tetrigoidea of the sampling, and Tetrigidae located at the end of the phylogeny. However, according to the research of *Chen (2005)* and *Yao (2008)*, Batrachididae was located at a more basal position and sistered to all other Tetrigoidea.

Animal mitogenome sequencing has exploded in recent years, and over 40,000 mitogenomes are avalialbe in the NCBI database (*Tan et al., 2017*). The insect mitogenome is typically a small, double-stranded circular molecule that ranges in size from 14 to 19 kb and encodes 37 genes (*Kim et al., 2005*). The mitogenome is one of the most extensively studied genomic systems and a widely used molecular component in the phylogenetic analysis of insects (*Cameron, 2014*), such as *Tarragoilus diuturnus* (*Zhou, Shi & Zhao, 2014*) and *Lerema accius* (*Cong & Grishin, 2016*).

Tetrigoidea is an important group in the phylogenetic and systemic studies of Caelifera; however, few complete mitogenomes were found in the GenBank database. Thus, currently, the phylogeny of Tetrigoidea is almost completely unknown (*Song et al., 2015*). For further study of the phylogenetic relationships among Tetrigoidea, the mitogenomes of *Formosatettix qinlingensis*, *Coptotettix longjiangensis* and *Thoradonta obtusilobata* were determined in this study. The phylogenetic analysis based on mitogenome data will provide a new insight for better understanding the phylogenetic relationship of Caelifera as well as Tetrigoidea.

## MATERIALS AND METHODS
### Sample collection and DNA extraction
Specimens of *F. qinlingensis* were collected in Shaanxi, China, those of *C. longjiangensis* in the Guangxi Zhuang Autonomous Region, China, and those of *T. obtusilobata* in Guizhou, China. Insects were preserved in 100% ethanol and stored at 4 °C. The total genomic DNA was extracted using the standard phenol/chloroform method (*Sambrook & Russell, 2001*).

## PCR amplification and sequencing by primer walking

Ten primary pairs of primers (Table S1) were used to amplify contiguous and overlapping fragments of the mitogenomes of *F. qinlingensis*, *C. longjiangensis* and *T. obtusilobata*, based on other published primer pairs (*Zhou, 2008*; *Simon et al., 2006*). PCR was performed in a total volume of 25 μL containing 12.5 μL of r-Taq mix (TaKaRa, Dalian, China), 9.5 μL of ddH$_2$O, 1 μL of each primer (10 μmol), and 1 μL of template DNA. The amplifications were performed under the following conditions: predenaturation at 96 °C for 2 min followed by 40 cycles of 96 °C for 20 s, 50.4 °C for 90 s and 68 °C for 4 min and a final extension at 72 °C for 7 min. PCR products were sequenced by Beijing Huada Gene Technology Co., LTD.

## Sequence assembly, annotation and analysis

The mitogenome sequences of *F. qinlingensis*, *C. longjiangensis* and *T. obtusilobata* were assembled using the Staden package 1.7.0 (*Staden, Beal & Bonfield, 2000*). Most of the transfer RNAs were identified by tRNAscan-SE 1.21 (*Lowe & Eddy, 1997*), and the other genes were determined by comparison with *T. japonica* (GenBank accession number JQ340002). The secondary structures of rRNA were inferred by comparison with those of *Pedopodisma emiensis* (*Zeng, 2014*) and *Gomphocerus sibiricus* (*Zhang, 2013*).

The nucleotide base compositions were calculated with Geneious 10.1.3 (*Kearse et al., 2012*), while the relative synonymous codon usage (RSCU) values for PCGs were calculated using MEGA 6.0 (*Tamura et al., 2013*). Composition skew analysis was conducted with formulas AT-skew=[A−T]/[A + T] and GC-skew=[G −C]/[G + C] (*Perna & Kocher, 1995*). The nonsynonymous substitution rate (Ka) and the synonymous substitution rate (Ks) were analyzed in DnaSP5.1 (*Librado & Rozas, 2009*).

## Phylogenetic analyses

In this study, the complete mitogenomes of 21 members of Caelifera, including three newly determined sequences of *F. qinlingensis*, *C. longjiangensis* and *T. obtusilobata* were used in the phylogenetic analysis (Table S2). Three species of Ensifera were used as the out-groups (Table S2). Thirteen protein-coding genes (PCG) and two rRNA genes were used for the construction of phylogenetic trees. All PCGs were aligned at the amino acid level using the default settings in MEGA 6.0 (*Tamura et al., 2013*), and the alignments were back translated to the corresponding nucleotide sequences. Because of high variability, the stop codons in PCGs were excluded in the alignment (*Zhang et al., 2014*; *Shuang-Shuang et al., 2014*). Two rRNA genes were aligned using Clustal X1.83 (*Thompson et al., 1997*), respectively. Finally, a PCG12 data set of 7,580 bp containing the first and second codon sites of 13 PCGs, a PCG123RY data set of 11,370 bp containing 13 PCGs with the third codon sites employing RY-coding strategy, a PCG12rRNA data set of 9,950 bp containing the first and second codon sites of 13 PCGs and two rRNA genes, and a PCG123RYrRNA data set of 13,740 bp containing 13 PCGs with the third codon sites employing RY-coding strategy and two rRNA genes were used for the phylogenetic analyses. PartitionFinder v1.1.1 (*Lanfear et al., 2012*) was used to search the optimal partitions and best models, with the "unlinked" branch lengths, "BIC" model selection, and "greedy" algorithm (Table 1).
**Table 1  The optimal partitions and best models for different data sets selected by using PartitionFinder v1.1.1.**

| Dataset | Partition | Optimal partitions | Best model |
|---|---|---|---|
| PCG12-ML | P1 | atp8_pos1, nad2_pos1, nad6_pos1 | GTR + I + G |
| | P2 | atp6_pos1, cox1_pos1, cox2_pos1, cox3_pos1, cytb_pos1, nad3_pos1 | GTR + I + G |
| | P3 | nad1_pos1, nad4L_pos1, nad4_pos1, nad5_pos1 | GTR + I + G |
| | P4 | atp6_pos2, atp8_pos2, cox1_pos2, cox2_pos2, cox3_pos2, cytb_pos2, nad2_pos2, nad3_pos2, nad6_pos2 | GTR + I + G |
| | P5 | nad1_pos2, nad4L_pos2, nad4_pos2, nad5_pos2 | GTR + I + G |
| PCG12-BI | P1 | atp8_pos1, nad2_pos1, nad6_pos1 | GTR + I + G |
| | P2 | atp6_pos1, cox1_pos1, cox2_pos1, cox3_pos1, cytb_pos1, nad3_pos1 | GTR + I + G |
| | P3 | nad1_pos1, nad4L_pos1, nad4_pos1, nad5_pos1 | GTR + I + G |
| | P4 | atp6_pos2, atp8_pos2, cox1_pos2, cox2_pos2, cox3_pos2, cytb_pos2, nad2_pos2, nad3_pos2, nad6_pos2 | GTR + I + G |
| | P5 | nad1_pos2, nad4L_pos2, nad4_pos2, nad5_pos2 | GTR + I + G |
| PCG123RY-ML | P1 | atp8_pos1, nad2_pos1, nad6_pos1, nad6_pos3 | GTR + G |
| | P2 | atp6_pos1, cox1_pos1, cox2_pos1, cox3_pos1, cytb_pos1, nad3_pos1 | GTR + I + G |
| | P3 | nad1_pos1, nad4L_pos1, nad4_pos1, nad5_pos1 | GTR + I + G |
| | P4 | atp6_pos2, atp8_pos2, cox1_pos2, cox2_pos2, cox3_pos2, cytb_pos2, nad2_pos2, nad3_pos2, nad6_pos2 | GTR + I + G |
| | P5 | nad1_pos2, nad4L_pos2, nad4_pos2, nad5_pos2 | GTR + I + G |
| | P6 | atp6_pos3, atp8_pos3, cox1_pos3, cox2_pos3, cox3_pos3, cytb_pos3, nad1_pos3, nad2_pos3, nad3_pos3, nad4L_pos3, nad4_pos3, nad5_pos3 | GTR + G |
| PCG123RY-BI | P1 | atp8_pos1, atp8_pos2, atp8_pos3, nad1_pos3, nad2_pos1, nad2_pos3, nad4L_pos3, nad4_pos3, nad5_pos3, nad6_pos1, nad6_pos3 | GTR + G |
| | P2 | atp6_pos1, cox1_pos1, cox2_pos1, cox3_pos1, cytb_pos1, nad3_pos1 | GTR + I + G |
| | P3 | nad1_pos1, nad4L_pos1, nad4_pos1, nad5_pos1 | GTR + I + G |
| | P4 | atp6_pos2, cox1_pos2, cox2_pos2, cox3_pos2, cytb_pos2, nad2_pos2, nad3_pos2, nad6_pos2 | GTR + I + G |
| | P5 | nad1_pos2, nad4L_pos2, nad4_pos2, nad5_pos2 | GTR + I + G |
| | P6 | atp6_pos3, cox1_pos3, cox2_pos3, cox3_pos3, cytb_pos3, nad3_pos3 | SYM + G |
| PCG12 + rRNA-ML | P1 | atp8_pos1, nad2_pos1, nad6_pos1 | GTR + I + G |
| | P2 | atp6_pos1, cox1_pos1, cox2_pos1, cox3_pos1, cytb_pos1, nad3_pos1 | GTR + I + G |
| | P3 | nad1_pos1, nad4L_pos1, nad4_pos1, nad5_pos1, rrnL, rrnS | GTR + I + G |
| | P4 | atp6_pos2, atp8_pos2, cox1_pos2, cox2_pos2, cox3_pos2, cytb_pos2, nad2_pos2, nad3_pos2, nad6_pos2 | GTR + I + G |
| | P5 | nad1_pos2, nad4L_pos2, nad4_pos2, nad5_pos2 | GTR + I + G |

*(continued on next page)*

**Table 1** (*continued*)

| Dataset | Partition | Optimal partitions | Best model |
|---|---|---|---|
| PCG12 + rRNA-BI | P1 | atp8_pos1, atp8_pos2, nad2_pos1, nad6_pos1 | GTR + G |
| | P2 | atp6_pos1, cox1_pos1, cox2_pos1, cox3_pos1, cytb_pos1, nad3_pos1 | GTR + I + G |
| | P3 | nad1_pos1, nad4L_pos1, nad4_pos1, nad5_pos1, rrnL, rrnS | GTR + I + G |
| | P4 | atp6_pos2, cox1_pos2, cox2_pos2, cox3_pos2, cytb_pos2, nad2_pos2, nad3_pos2, nad6_pos2 | GTR + I + G |
| | P5 | nad1_pos2, nad4L_pos2, nad4_pos2, nad5_pos2 | GTR + I + G |
| PCG123RY + rRNA-ML | P1 | atp6_pos3, atp8_pos1, atp8_pos3, cox1_pos3, cox2_pos3, cox3_pos3, cytb_pos3, nad1_pos3, nad2_pos1, nad2_pos3, nad3_pos3, nad4L_pos3, nad4_pos3, nad5_pos3, nad6_pos1, nad6_pos3 | GTR + G |
| | P2 | atp6_pos1, cox1_pos1, cox2_pos1, cox3_pos1, cytb_pos1, nad3_pos1 | GTR + I + G |
| | P3 | nad1_pos1, nad4L_pos1, nad4_pos1, nad5_pos1, rrnL, rrnS | GTR + I + G |
| | P4 | atp6_pos2, atp8_pos2, cox1_pos2, cox2_pos2, cox3_pos2, cytb_pos2, nad2_pos2, nad3_pos2, nad6_pos2 | GTR + I + G |
| | P5 | nad1_pos2, nad4L_pos2, nad4_pos2, nad5_pos2 | GTR + I + G |
| PCG123RY + rRNA-BI | P1 | atp8_pos1, atp8_pos2, atp8_pos3, nad1_pos3, nad2_pos1, nad2_pos3, nad3_pos3, nad4L_pos3, nad4_pos3, nad5_pos3, nad6_pos1, nad6_pos3 | GTR + G |
| | P2 | atp6_pos1, cox1_pos1, cox2_pos1, cox3_pos1, cytb_pos1, nad3_pos1 | GTR + I + G |
| | P3 | nad1_pos1, nad4L_pos1, nad4_pos1, nad5_pos1, rrnL, rrnS | GTR + I + G |
| | P4 | atp6_pos2, cox1_pos2, cox2_pos2, cox3_pos2, cytb_pos2, nad2_pos2, nad3_pos2, nad6_pos2 | GTR + I + G |
| | P5 | nad1_pos2, nad4L_pos2, nad4_pos2, nad5_pos2 | GTR + I + G |
| | P6 | atp6_pos3, cox1_pos3, cox2_pos3, cox3_pos3, cytb_pos3 | SYM + G |

**Notes.**

pos1, the first codon site of each PCG; pos2, the second codon site of each PCG; pos3, the third codon site of each PCG.

The phylogenies were determined using both maximum likelihood (ML) and Bayesian inference (BI) methods. The ML analysis was performed using the program RAxML version 7.0.3 (*Stamatakis, 2006*), and the optimal partitions and best models were selected by using PartitionFinder v1.1.1 (*Lanfear et al., 2012*). A bootstrap analysis was performed with 1,000 replicates. The BI analysis was performed using MrBayes version 3.1.2 (*Ronquist & Huelsenbeck, 2003*), and also employing the optimal partitions and best models selected by PartitionFinder v1.1.1 (*Lanfear et al., 2012*). According to Markov Chain Monte Carlo analysis, four chains (one cold and three heated chains) were set to run simultaneously for 1,000,000 generations. Each set was sampled every 100 generations with a burn-in of 25%, and the remaining samples were used to obtain the consensus tree. The effective sample size (ESS) values were analyzed by Tracer v1.5 (*Rambaut, Suchard & Drummond, 2004*), with ESS values greater than 200.

## RESULTS AND DISCUSSION

### Mitochondrial genomic structure

The size of the mitogenome sequence obtained from *F. qinlingensis*, *C. longjiangensis* and *T. obtusilobata* was 15,180, 14,495 and 14,538 bp, respectively (Table 2). The three mitogenomes were deposited in the GenBank database under accession numbers KY798412 (*F. qinlingensis*), KY798413 (*C. longjiangensis*) and KY798414 (*T. obtusilobata*). The gene composition, order, and orientation of all three mitogenomes were the same as those of the mitogenomes of other Tetrigoidea species, such as *T. japonica* (JQ340002), and each sequence included 13 PCGs, partial sequences of rRNA genes (rRNAs), tRNA genes (tRNAs) and a A + T-rich region (Table 2; Fig. 1). As shown in other Tetrigoidea species, transcribed from the light strand were two rRNAs, four PCGs and eight tRNAs (Table 2). The A + T contents were 75.6%, 73.1% and 71.8% in the mitogenomes of the Tetrigoidea species *F. qinlingensis*, *C. longjiangensis* and *T. obtusilobata*, respectively.

### Nucleotide composition and skew

A comparative analysis of A + T content vs AT-skew and G + C content vs GC-skew within Caelifera mitogenomes is shown in Fig. 2. The approximately positive correlations were found between A + T content and AT-skew, and as well as between G + C content and GC-skew (Figs. 2A and 2B). The trends of increased A + T content and AT-skew were roughly Tridactyloidea < Eumastacoidea < Acridoidea/Tetrigoidea, while the increased G + C content and GC-skew were roughly Acridoidea/Tetrigoidea < Tridactyloidea.

The average AT-skew of Caelifera mitogenomes was 0.15, ranging from 0.01 in *Ellipes minuta* to 0.22 in *C. longjiangensis* (Table S3). The average GC-skew of mitogenomes was −0.19, ranging from −0.30 in *E. minuta* to −0.11 in *Pielomastax zhengi* (Table S3). The Tridactyloidea had lower A + T content and A-skew, higher G + C content and C-skew compared with other superfamily in Caelifera.

### Spacers and overlaps

A total of seven intergenic spacers ranging from 1 to 12 bp were found in the mitogenome of *F. qinlingensis*. Among these spacers, the longest noncoding region (12 bp) was found between trnS(UCN) and nad1. Overlapping regions ranging from 1 to 8 bp occurred in the *F. qinlingensis* mitogenome, such as the 8 bp overlap between trnW and trnC. Most of the intergenic spacers and overlapping regions in *F. qinlingensis* were similar to those in the mitogenomes of the other two species of Tetrigoidea. However, a long intergenic spacer occurred between trnS(UCN) and nad1 in *C. longjiangensis* (131 bp) and *T. obtusilobata* (399 bp). Long noncoding regions between trnS(UCN) and nad1 also occur in the insect orders Hymenoptera, Coleoptera and Hemiptera, and in other orthopterans, with a length from 40 to 300 bp. For examfigur 2le, *Xyleus modestus* (Orthoptera: Caelifera) contains a noncoding region (259 bp) between trnS(UCN) and nad1 (*Sheffield et al., 2010*). Moreover, some conserved sequences occur, such as ATACTAA in Lepidoptera, TACTA in Coleoptera, and THACWW in Hymenoptera (*Wei, 2009*). However, although the sequences in Orthoptera had low similarity, sequence conservation was observed at the superfamily level (Figs. 3A and 3D). Sequences (TTCTAWTTTT) in Tetrigoidea and
**Table 2** Annotation of the mitochondrial genomes of *Formosatettix qinlingensis* (F. q), *Coptotettix longjiangensis* (C. l) and *Thoradonta obtusilobata* (T. o).

| Feature | Strand | Position | | | Initiation codon/Stop codon | | |
|---|---|---|---|---|---|---|---|
| | | F. q | C. l | T. o | F. q | C. l | T. o |
| trnI | J | <1–54 | <1–25 | | | | |
| trnQ | N | 55–123 | 27–95 | | | | |
| trnM | J | 123–191 | 95–163 | <1–17 | | | |
| nad2 | J | 192–1,193 | 164–1,165 | 18–1,028 | ATG/TAA | GTG/TAA | ATT/TAA |
| trnW | J | 1,192–1,257 | 1,169–1,234 | 1,027–1,092 | | | |
| trnC | N | 1,250–1,315 | 1,227–1,291 | 1,085–1,146 | | | |
| trnY | N | 1,316–1,379 | 1,294–1,358 | 1,147–1,212 | | | |
| cox1 | J | 1,377–2,915 | 1,356–2,894 | 1,210–2,748 | ATC/TAA | ATC/TAA | ATC/TAG |
| trnL(uur) | J | 2,911–2,974 | 2,890–2,953 | 2,744-2,806 | | | |
| cox2 | J | 2,975–3,658 | 2,954–3,637 | 2,807–3,484 | ATG/TAA | ATG/TAA | ATG/TAA |
| trnD | J | 3,657–3,719 | 3,636–3,700 | 3,483–3,545 | | | |
| trnK | J | 3,720–3,787 | 3,701–3,772 | 3,546–3,611 | | | |
| atp8 | J | 3,792–3,947 | 3,776–3,934 | 3,612–3,764 | ATG/TAA | ATG/TAA | ATG/TAA |
| atp6 | J | 3,941–4,612 | 3,934-4,605 | 3,758–4,429 | ATG/TAA | ATG/TAA | ATG/TAA |
| cox3 | J | 4,612–5,401 | 4,605–5,394 | 4,429–5,218 | ATG/T | ATG/T | ATG/T |
| trnG | J | 5,402–5,464 | 5,396–5,461 | 5,220–5,281 | | | |
| nad3 | J | 5,462–5,818 | 5,459–5,815 | 5,279–5,635 | ATT/TAG | ATA/TAG | ATA/TAG |
| trnA | J | 5,817–5,881 | 5,814–5,878 | 5,634–5,696 | | | |
| trnR | J | 5,881–5,943 | 5,878–5,942 | 5,696–5,758 | | | |
| trnN | J | 5,940–6,003 | 5,939–6,002 | 5,751–5,814 | | | |
| trnS(agn) | J | 6,003–6,071 | 6,002–6,070 | 5,814–5,882 | | | |
| trnE | J | 6,071–6,134 | 6,070–6,132 | 5,882–5,944 | | | |
| trnF | N | 6,133–6,195 | 6,131–6,193 | 5,943–6,005 | | | |
| nad5 | N | 6,199–7,915 | 6,194–7,910 | 6,009–7,722 | ATG/T | ATG/T | ATG/T |
| trnH | N | 7,919–7,982 | 7,914–7,977 | 7,724–7,785 | | | |
| nad4 | N | 7,982–9,307 | 7,977–9,302 | 7,785–9,110 | ATG/TAG | ATG/TAG | ATG/TAG |
| nad4L | N | 9,301–9,591 | 9,296–9,586 | 9,104–9,388 | ATT/TAA | ATT/TAA | ATT/TAA |
| trnT | J | 9,594–9,658 | 9,589–9,653 | 9,391–9,452 | | | |
| trnP | N | 9,659–9,722 | 9,654–9,717 | 9,453–9,517 | | | |
| nad6 | J | 9,724–10,218 | 9,719–10,216 | 9,519–10,013 | ATG/TAA | ATG/TAA | TTG/TAA |
| cytb | J | 10,218–11,354 | 10,216–11,352 | 10,013–11,149 | ATG/TAG | ATG/TAA | ATG/TAA |
| trnS(ucn) | J | 11,353–11,420 | 11,366–11,433 | 11,148-11,214 | | | |
| nad1 | N | 11,433–12,377 | 11,565–12,509 | 11,614–12,564 | ATA/TAA | GTA/TAA | ACA/TAA |
| trnL(cun) | N | 12,372–12,434 | 12,504–12,565 | 12,559–12,623 | | | |
| rrnL | N | 12,435–13,726 | 12,566–13,858 | 12,625–13,909 | | | |
| trnV | N | 13,728–13,799 | 13,861–13,932 | 13,910–13,980 | | | |
| rrnS | N | 13,800–14,580 | 13,933–>14,495 | 13,981–>14,538 | | | |
| A + T-rich region | | 14,581–~15,180 | | | | | |

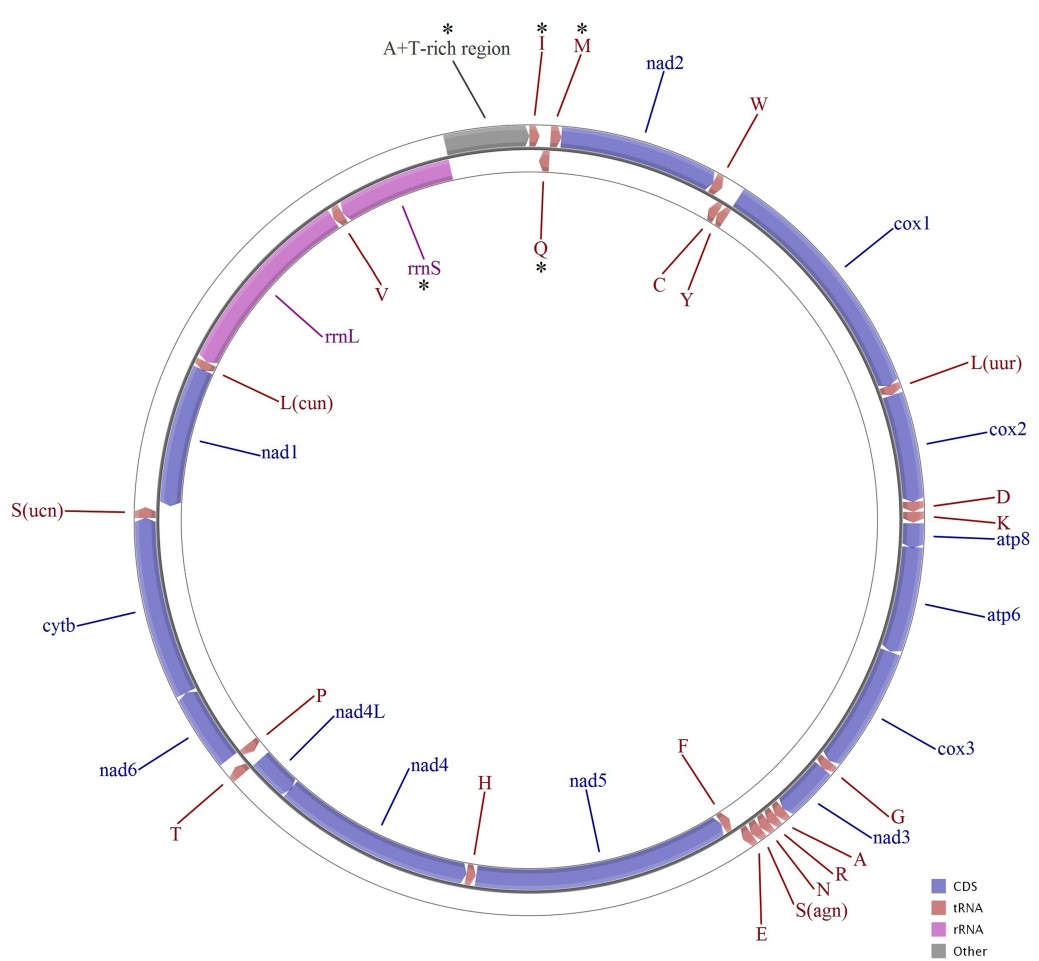

**Figure 1 Mitochondrial map of three Tetrigoidea species (*Formosatettix qinlingensis*, *Coptotettix longjiangensis* and *Thoradonta obtusilobata*).** Note: * means partial or not sequenced genes.

sequences (TTCTNRAAA) in Acridoidea were conserved (Figs. 3A and 3B); therefore, these conserved sequences might be useful for the identification of Tetrigoidea and Acridoidea.

## Protein-coding genes

In *F. qinlingensis*, *C. longjiangensis* and *T. obtusilobata*, the A + T content of PCGs was 74.7%, 72.0% and 70.5%, respectively. For each PCG of the three Tetrigoidea mitogenomes, the A + T contents of atp8 and nad6 were much higher and those of COX genes in all three species lower than those of the other genes (Fig. S1), which are similar results to those found by *Zhang et al. (2013b)*. Four PCGs (nad5, nad4, nad4L and nad1) coded by the N-strand had a T-skewed value, whereas each PCG in the J-strand was C-skewed, and each PCG in the N-strand was G-skewed (Fig. S1), which are results similar to those for Gomphocerinae mitogenomes (*Zhang et al., 2013b*).

For the initial and termination codons, the most common start codon was ATG. Start codons GTG, ATT, ATC, ATA, GTA and ACA also occurred in the Tetrigoidea species, with some of them conserved, such as ATC in cox1. The same use of ATC in cox1 is found in other

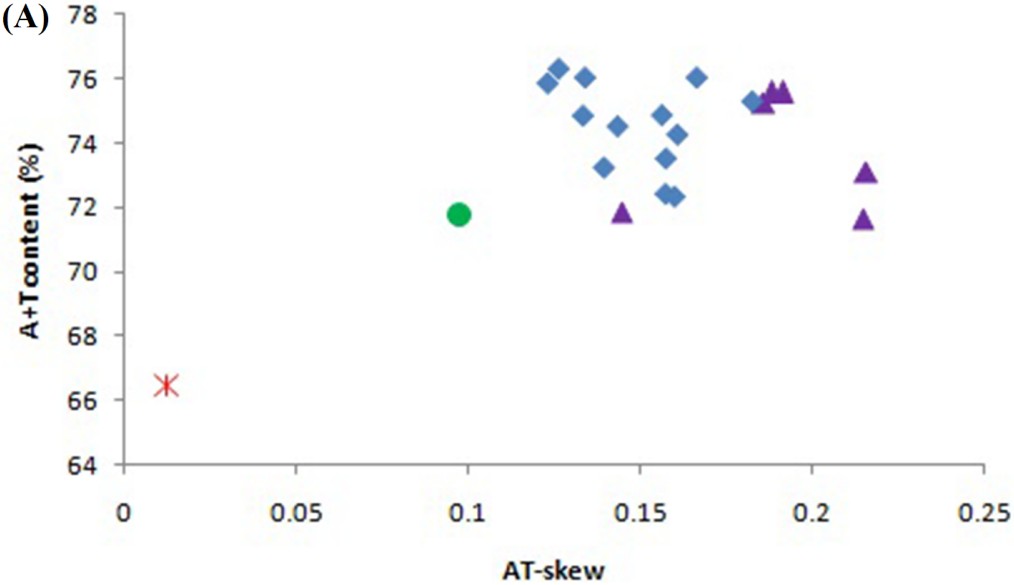

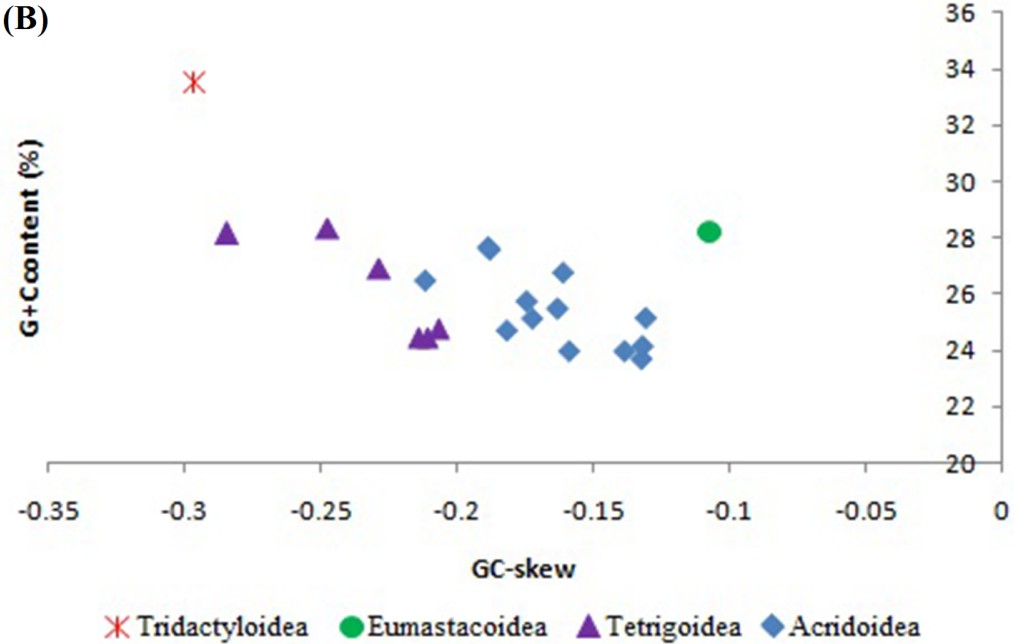

**Figure 2** **The A + T content vs AT-skew and G + C content vs GC-skew in Caelifera mitogenomes.** (A) A+T content vs AT-skew; (B) G+C content vs GC-skew.

Caelifera, such as *Calliptamus italicus* (EU938373), *Oxya chinensis* (EF437157), *Prumna arctica* (GU294758) and *Traulia szetschuanensis* (EU914849) of Acridoidea, *P. zhengi* (JF411955) of Eumastacoidea, and *A. yunnanensis* (JQ272702) and *T. japonica* (JQ340002) of Tetrigoidea.

## (A) Tetrigoidea

| | |
|---|---|
| *Alulatettix yunnanensis* | ------TTCTATTTTTTA- |
| *Trachytettix bufo* | ----CTTTTCTAATTTTTAT |
| *Formosatettix qinlingensis* | ------TTCTATTTTTTA- |
| *Coptotettix longjiangensis* | ---ATTTTCTATTTTTTA- |
| *Tetrix japonica* | ------TTCTATTTTTTA- |
| *Thoradonta obtusilobata* | AATTATTTCTAATTTTATT |

## (B) Acridoidea

| | |
|---|---|
| *Acrida cinerea* | -----TAAAATTCTAAAAAAAATTAAC |
| *Ceracris kiangsu* | ---------GTTCTAAAAATAATTAA- |
| *Oxya chinensis* | ---TTCTAT--TCTAAAAAAATTTAA- |
| *Mekongiella xizangensis* | ---TAGTTATTCTTAAAAAATTTCA- |
| *Filchnerella helanshanensis* | CTATTGTATTTCTGAAAAAATTTCA- |
| *Pseudotmethis rubimarginis* | CTATTGTATTTCTGAAAAAATTTCA- |
| *Calliptamus italicus* | -----TTTAATTCTCAAAAAATTTCA- |
| *Atractomorpha sinensis* | ---------TTCTCAAAAAATTTCA- |
| *Traulia szetschuanensis* | -----GTAAATTCTTAAAAAATTTCA- |
| *Arcyptera coreana* | -----ATAAATTCTAAAAAAATTTAA- |
| *Locusta migratoria* | -----TTAAATTCTTAAA--ATTTAA- |
| *Prumna arctica* | -----ATTAATTCTAGAAAAATTTCA- |
| *Gomphocerus sibiricus* | -----ATTATTTCTAGAAAAATTTCA- |

## (C) Eumastacoidea

| | |
|---|---|
| *Pielomastax zhengi* | AATTGTTCTTGTTTTATTTGA |

## (D) Tridactyloidea

| | |
|---|---|
| *Ellipes minuta* | TGTACAAAATTTATTTCA |

**Figure 3** **Alignments of the intergenic spacer between trnS(UCN) and nad1 genes in caeliferan mitogenomes.** (A) Tetrigoidea; (B) Acridoidea; (C) Eumastacoidea; (D) Tridactyloidea.

For all three Tetrigoidea species, stop codon usage was consistent in 11 PCGs (nad2, cox2, atp8, atp6, cox3, nad3, nad5, nad4, nad4L, nad6 and nad1). Cox3 and nad5 were terminated with the incomplete stop codon T in the three Tetrigoidea species. The terminal T serves as a stop signal after it is completed to UAA via post-transcriptional polyadenylation (*Ojala, Montoya & Attardi, 1981*).

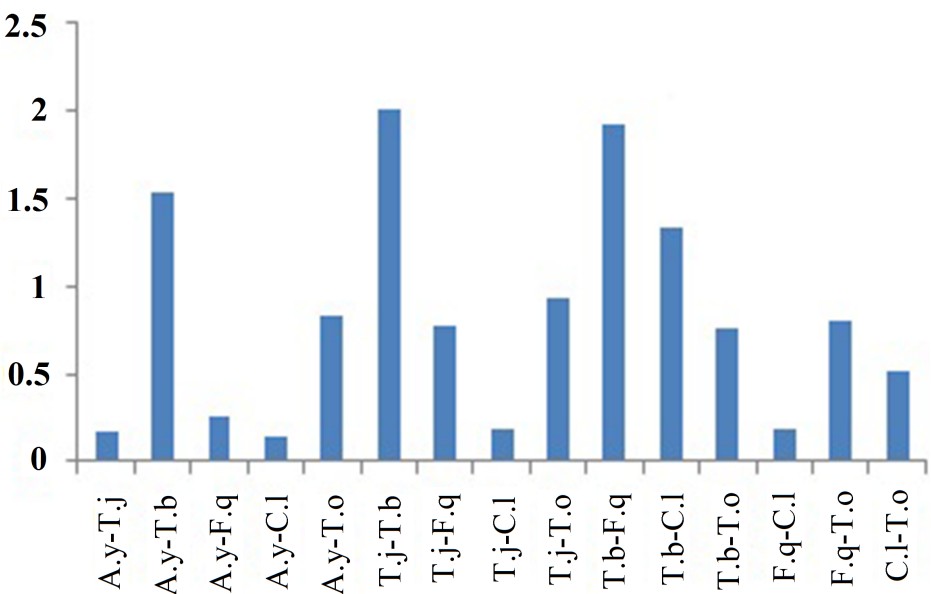

**Figure 4 The Ka/Ks values of atp8 gene with paired comparison in six Tetrigidae mitogenomes.** Note: A.y, *Alulatettix yunnanensis*; T.j, *Tetrix japonica*; T.b, *Trachytettix bufo*; F.q, *Formosatettix qinlingensis*; C.l, *Coptotettix longjiangensis*; T.o, *Thoradonta obtusilobata*.

The relative synonymous codon usage of Caelifera was analyzed. The use of the anticodons NNA and NNU was relatively frequent, while NNG and NNC was lower (Table S4). This result revealed the preference for A or T in the third position, which was similar to the results of whiteflies (*Chen et al., 2016*). Mitogenome encoded 22 tRNA genes, which were used to synthesis 20 amino acids. Some mostly used synonymous codons of PCGs did not correspond to the tRNA anticodons of mitogenomes. For example, UUU is the mostly used synonymous codon of Phe(F) (Table S4), while anticodon of trnF in the mitogenomes is UUC (Fig. S4). This result shows that the protein synthesis of mitogenomes not only depends on mitochondria encoded tRNAs, but also needs nuclear encoded tRNAs.

The average ratio of Ka/Ks was calculated for each PCG of six Tetrigidae mitogenomes. The results showed that atp8 had the highest evolutionary rate, while cox1 was the lowest (Table S5). The average ratios of Ka/Ks for each PCG were all below 1 (Table S5), indicating the existence of purifying selection. A roughly negative correlation was observed between the average ratio of Ka/Ks and the G + C content of each PCG (Table S5), which was also found in true bug mitogenomes (*Li et al., 2012*). The evolutionary patterns of mitochondrial genes were probably caused by the varied G + C content (*Hua et al., 2008*). Furthermore, the ratios of Ka/Ks for atp8 gene were above 1 in some pairwise comparison (Fig. 4), indicating under positive selection. The varied sites of atp8 gene might be associated with the evolution of *T. bufo* (Fig. 5).

## Ribosomal and transfer RNA genes

As in most insect mitogenomes, two rRNA genes (rrnL and rrnS) occurred in the three Tetrigoidea mitogenomes between trnL(cun) and the A + T-rich region, separated by

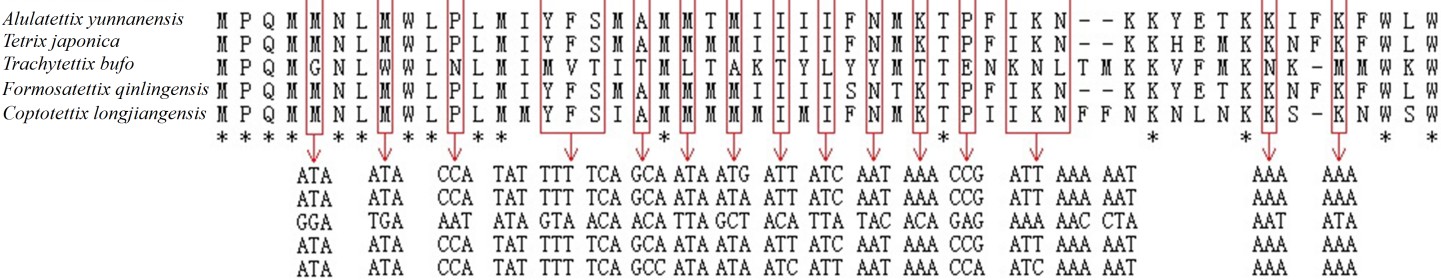

**Figure 5** The varied amino acid and corresponding nucleotide sequences of atp8 gene in five Tetrigidae mitogenomes.

trnV. The lengths of rrnS and rrnL determined in *F. qinlingensis* were 781 and 1,292 bp, respectively, and the A + T content of rrnS and rrnL was 76.7% and 79.3%, respectively.

The overall rrnS structure of *F. qinlingensis* included three domains (Fig. S2), which were identical with those predicted for other Caelifera species, such as *G. sibiricus* (*Zhang et al., 2013b*). The secondary structure of rrnL in *F. qinlingensis* contained six domains with domain III degenerated to a single strand as one bond (Fig. S3), which is a structure similar to that found in the study of *Zhang et al. (2013b)*. The percentage of conserved sites in the six domains among the three Tetrigoidea species showed that more conserved sites were in domains IV, V and VI than in other domains, whereas domain III had more variable sites.

A total of 22 tRNAs were found interspersed in the mitogenomes of *F. qinlingensis* and *C. longjiangensis*, which ranged in size from 54 bp (trnI) to 72 bp (trnV). Both trnL and trnS had two copies in the mitogenomes. Most of the tRNAs could be folded into the canonical cloverleaf secondary structure, except for trnS(agn) (Fig. S4). The trnS(agn) lacked the DHU arm in the three Tetrigoidea mitogenomes, which is a feature commonly observed in other Caelifera species (*Zhao et al., 2010*; *Liu & Huang, 2010*). Twenty-two non-Watson-Crick pairings were identified in tRNA genes of the *F. qinlingensis* mitogenome, including 18 G-U mismatches. Most of these G–U pairs were found in tRNAs on the N-strand. By contrast, in the study of *Asakawa et al. (1991)*, G–U pairs are found more frequently in tRNAs of the J-strand in mitogenomes of the various animals they examined. Two A–G pairs were predicted in the acceptor arm of trnW and trnR; one A–A pair was predicted in the acceptor arm of trnQ; and one C-U pair was predicted in the TψC arm of trnH (Fig. S4).

## A + T-rich region

A 600 bp A + T-rich region was observed between rrnS and trnI in the mitogenome of *F. qinlingensis*, which was composed of 80.8% A + T. The high mutation rate of this region might be related to the high A + T content and low selection pressure (*Yang et al., 2011*). *F. qinlingensis* had a larger A + T-rich region than that of other species of Tetrigoidea, e.g., 460 bp in *A. yunnanensis* (JQ272702) and 531 bp in *T. japonica* (JQ340002). Conserved or variable sections are not observed in the A + T-rich regions of insects; whereas tandem repetitions and conserved structural elements have been observed (*Zhang, Szymura & Hewitt, 1995*; *Zhang & Hewitt, 1997*). The A + T-rich region of *F. qinlingensis* contained

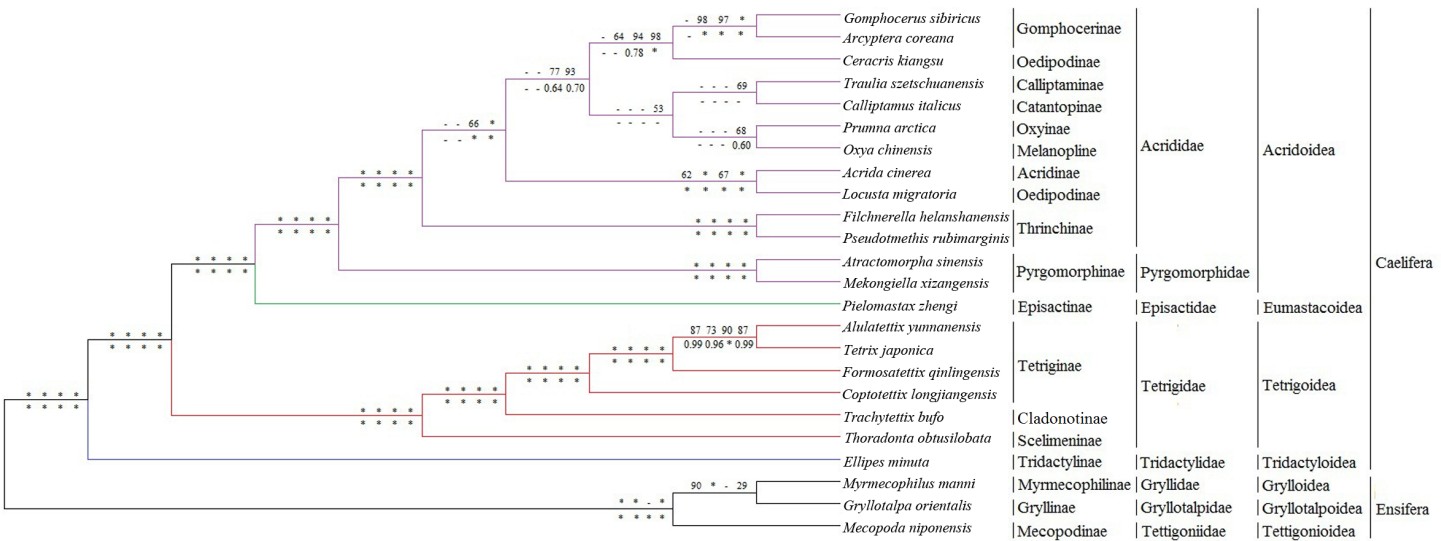

**Figure 6 Phylogenetic reconstructions of some Caelifera species based on different datasets and methods.** Node supports from left to right above lines are the results of ML trees of PCG12, PCG123RY, PCG12rRNA and PCG123RYrRNA datasets, under lines are BI trees of PCG12, PCG123RY, PCG12rRNA and PCG123RYrRNA datasets, respectively. *, bootstrap support of 100 in ML trees or Bayesian posterial probability of 1.00 in BI trees. -, no support for the clade.

tandem repeated sequences, and the repeats with (AATAATAAAAAAA)n ($n = 3.1$) were found at the 5′ end of the A + T-rich region (nt 30–71), with more A nucleotides.

## Phylogenetic analyses

The phylogenetic trees resulting from the PCG-ML and PCG-BI analyses were consistent, except for *Myrmecophilus manni* (Fig. 6 and Fig. S5). The ML and BI topologies of mitochondrial datasets generated similar tree topologies (Fig. 6 and Fig. S5).

The results of the phylogenetic relationships among the major superfamilies were largely congruent with previous studies (*Flook, Rowell & Gellissen, 1995*; *Leavitt et al., 2013*; *Song et al., 2015*). The relationships among four superfamilies of Caelifera were (((Acridoidea +Eumastacoidea) +Tetrigoidea) +Tridactyloidea), which is similar to the superfamily relationships determined in previous studies that used morphological and molecular evidence (*Flook, Rowell & Gellissen, 1995*; *Leavitt et al., 2013*; *Song et al., 2015*). In this study, Tridactyloidea was the sister group to Caelifera, and Tetrigoidea was located at a relatively basal position in Caelifera compared with Acridoidea and Eumastacoidea, which are relations consistent with those in the studies of *Flook & Rowell (1997b)* and *Song et al. (2015)*. The results strongly supported the monophyly of Tetriginae, sister to the Cladonotinae, whereas Scelimeninae was in the basal position. The relationships among Tetrididae were (((((*Alulatettix yunnanensis* + *Tetrix japonica*) + *Formosatettix qinlingensis*) + *Coptotettix longjiangensis*) + *Trachytettix bufo*) + *Thoradonta obtusilobata*).

In this study, Acrididae was the sister group of Pyrgomorphidae in Acridoidea. The phylogenetic relationships of subfamilies in Acrididae were (((((Gomphocerinae + Oedipodinae) + ((Calliptaminae + Catantopinae) + (Oxyinae + Melanoplinae))) + (Acridinae + Oedipodinae)) + Thrinchinae). However, the phylogenetic relationships

within Acrididae obtained in this study contained some differences with other studies (*Zhang et al., 2013a*), such as a clade including *A. cinerea* and *L. migratoria*, which might be caused by different sampling approaches. Apart from different sampling approaches, hybridization might be a major reason for the difference, as hybridization has been observed and described in a number of acridoid species (*Gottsberger, 2007*; *Hochkirch & Lemke, 2011*; *Rohde et al., 2015*).

# CONCLUSIONS

The mitogenomes of *Formosatettix qinlingensis*, *Coptotettix longjiangensis* and *Thoradonta obtusilobata* were sequenced in this study. The analyses of mitochondrial features showed that conserved sequences were observed in intergenic spacers at the superfamily level. The phylogenetic results support the relationship of (((((*Tetrix japonica*, *Alulatettix yunnanensis*), *Formosatettix qinlingensis*), *Coptotettix longjiangensis*), *Trachytettix bufo*), *Thoradonta obtusilobata*) in Tetrigoidea.

## Funding

This work was supported by the National Natural Science Foundation of China (Grant No. 31601846), Natural Science Foundation of Shaanxi Province, China (Grant No. 2017JQ3014), China Postdoctoral Science Foundation (Grant No. 2016M602760) and Fundamental Research Funds for the Central Universities, China (Grant no. GK201603109 and GK201603112). The funders had no role in study design, data collection and analysis, decision to publish, or preparation of the manuscript.

## Grant Disclosures

The following grant information was disclosed by the authors:
National Natural Science Foundation of China: 31601846.
Natural Science Foundation of Shaanxi Province, China: 2017JQ3014.
China Postdoctoral Science Foundation: 2016M602760.
Fundamental Research Funds for the Central Universities, China: GK201603109, GK201603112.

## Competing Interests

The authors declare there are no competing interests.

## Author Contributions

- Li-Liang Lin conceived and designed the experiments, performed the experiments, analyzed the data, contributed reagents/materials/analysis tools, wrote the paper.
- Xue-Juan Li performed the experiments, analyzed the data, wrote the paper, prepared figures and/or tables.
- Hong-Li Zhang analyzed the data.
- Zhe-Min Zheng conceived and designed the experiments, reviewed drafts of the paper.

## DNA Deposition

The following information was supplied regarding the deposition of DNA sequences:

The three mitogenomes were deposited in the GenBank database under accession numbers KY798412 (*F. qinlingensis*), KY798413 (*C. longjiangensis*) and KY798414 (*T. obtusilobata*).

## Data Availability

All analysis results are provided in the Supplemental Files.

## Supplemental Information

Supplemental information for this article can be found online at http://dx.doi.org/10.7717/peerj.4002#supplemental-information.

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
