# Peer review of "Mitochondrial genomes of three Tetrigoidea species and phylogeny of Tetrigoidea"

_PeerJ, doi:10.7717/peerj.4002_

## Round 0.1 · original submission · Minor Revisions

· Academic Editor

Minor Revisions

Please try to comply with reviewers questions, all of them I consider appropriate (including those in the annotated pdf).

I stress further the need for careful English revision (and a lot of typos too).

Finally, I consider it essential to address the limitations of mtDNA in phylogeny as well as the issue (also raised by reviewers) of hybridization in this context.

Reviewer 1 ·

Basic reporting

Clear, unambiguous, professional English language used throughout but I noticed a couple of things worth revising:

Lines 128-129 “The positive relativity were found between A+T content and AT-skew, and also between G+C content and GC-skew.”

Replace “mt genomes” with mitogenomes (e.g. in line 132).

The graphic resolution of the main figures should be improved in the final submission for publication.

Experimental design

No comment.

Validity of the findings

No comment.

Additional comments

This is a well-structured and well written work reporting the near-complete mitogenomes of three Tetrigoidea species. The paper is adequate in its scope, and fits well in the standards of PeerJ.

·

Basic reporting

Although the article is written in clear English, the authors should check again carefully for typos. Striking examples:
pdf line 37: "Batraehididae, Cladonltidae, Diseotettigidae, ..."
Fig. 3: "Tridactuloidea" instead of "Tridactyloidea"

The names of the species studied should be given in full at first mentioning with the genus not abbreviated.

Experimental design

pdf lines 30-38: The authors compare two systematic divisions of Tetrigidae (OSF versus Zheng) that - apart from the level subfamily versus family - is nearly identic except that two taxa are missing in the latter system. Is this due to different arrangement of species or only due to the two (sub-)families missing in China?

pdf lines 56-61: The exact purpose of the study, apart from adding new data, is not clear from the introduction. The authors state that they "further study of the phylogenetic relationships among Tetrigoidea". How will they reach this aim? Are the species selected representatives of different (sub-)families or do they represent (sub-)families, of which genome data are not yet available from genbank?

The aim of the study should given more precisely and in accordance with what is presented in results and discussion.

Validity of the findings

no comment

·

Basic reporting

The authors present the description of three mitogenomes of Orthoptera of the family Tetrigidae, basically in a data release manuscript. The manuscript is well written and easy to understand. The structure is adequate, as the results are rather descriptive and leave comparably little room for discussion.

Experimental design

The methods are suitably applied, largely well described, and the results are described in high detail.

Validity of the findings

Overall, the findings appear valid. My main point of criticism is the following:

Molecular phylogenetic studies of Caelifera are notorious for the difficulties presented by mitochondrial haplotype sharing (see references pp. 6/7 in Hawlitschek et al. 2016, http://onlinelibrary.wiley.com/doi/10.1111/1755-0998.12638/full). While most previous studies focus on Acrididae, not on Tetrigidae (or related families in Tetrigoidea), the authors should take into account that hybridization (which has been postulated also across genera) or other phenomena causing misleading results may also influence their Tetrigoidea results. The authors should introduce these problems and carefully discuss their implication for their results.

Additional comments

Other than that, I only have minor comments, a list of which is given below.

Kind regards,

Oliver Hawlitschek


l. 16: Replace "vaule" with "value".

l. 30 / 31: Add citations for the first two sentences of the introduction.

l. 39: "minor damage to crops": better write "minor importance as agricultural pests".

l. 47: "original taxon": does this mean "sister group to all other Tetrigoidea"?

l. 53: Replace "widely used molecular components" with "a widely used molecular component".

l. 69: "PCR amplification and sequencing by primer walking": Text should be bold.

l. 89 / 90: "18 members of Caelifera and F. qinlingensis, […]": But the species of Tetrigoidea studied here are members of Caelifera, too?

l. 128: "positive relativity": Do you mean "positive correlation"?

l. 132: I do not understand the importance of true bug mitogenomes here. If this is a standard method I do not know it. Please explain in the methods section.

l. 172-174: "Some highest […] encoded tRNAs.": These sentences are difficult to understand, please check the wording.

l. 224: "original taxon": See my comment to l. 47.

l. 230: Replace "Melanopline" with "Melanoplinae".

l. 237/238: "topologies of Tetrigoidea based on 16S rRNA": It is not clear where these results come from. Did the authors conduct specific analyses based on 16S only? This is not described in the methods section. Or are they from Eades et al. (2014)? Please clarify.

l. 270: Replace "Zhemin zheng" with "Zhemin Zheng".

l. 393: Replace "Potua Sabulosa" with "Potua sabulosa".

l. 397: Please give the website for the download of Tracer.

l. 406: Replace "Molecular evolution" with "molecular evolution".

l. 422: Italicize "Tetrix tenuicornis".

l. 440: Replace "Comparative & phylogentic" with "comparative & phylogenetic".

Reviewer 4 ·

Basic reporting

no comment

Experimental design

no comment

Validity of the findings

There is one aspect in the paper which I find speculative at the moment. I explained it in a comment in the attached pdf (line 150).

Additional comments

I added some comments and suggestions in the attached pdf.

Annotated reviews are not available for download in order to protect the identity of reviewers who chose to remain anonymous.

---

## Round 0.2 · Minor Revisions

· Academic Editor

Minor Revisions

Please address the remaining minor suggestions of the two reviewers.

Reviewer 1 ·

Basic reporting

Following on my previous comments:

Section 'Nucleotide composition and skew':
The phrase strating with 'The approximate positive correlation were found (...).' need revision. A-skew and C-skew should be replace by A+T-skew and G+C-skew for the sake of consistency.

All figures still need to be improved with regards to resolution for publication. For example, I found it difficult to read a printed version of the phylogenetic tree.

Experimental design

No comment.

Validity of the findings

No comment.

·

Basic reporting

no comment

Experimental design

no comment

Validity of the findings

no comment

·

Basic reporting

No further comments.

Experimental design

No further comments.

Validity of the findings

No further comments.

Additional comments

The authors have thoroughly revised their paper, improving it considerably. I only have a few minor details, mostly stylistic, that should be taken into account.

Kind regards

Oliver Hawlitschek


1. l. 33: "and one tribe": Xerophyllini is a tribe within Cladonotinae. Other subfamilies of Tetrigidae have more tribes. Best remove this.

2. l. 53: Change "Cong and Grishin., 2016" to "Cong and Grishin, 2016".

3. l. 59: Change "represent for Scelimeninae, make it possible" to "to represent Scelimeninae, making it possible".

4. l. 60/61: Change "Combined with other 18 Caelifera mitogenomes in the database, the comparative analysis was made" to "A comparative analysis with other 18 Caelifera mitogenomes was conducted". But see my next comment.

5. l. 54-66: Most of this entire paragraph is a summary of the methods section and therefore redundant. I suggest keeping only the first, second, and last sentence and removing the rest of the paragraph.

6. l. 156: Change "the identify of Tetrigoidea" to "the identification of Tetrigoidea".

7. l. 176: Change "NNU were relatively" to "NNU was relatively" and "NNC were lower" to "NNC was lower".

8. l. 178-182: "Some highest … tRNAs": This sentence is still not clear. Please rephrase again.

9. l. 245: Hawlitschek et al. (2016) is not the ideal reference here, as it does not really discuss hybridization. Check p. 2 of this paper for more suitable references (e.g., Gottsberger 2007, Hochkirch & Lemke 2011, Rohde et al. 2015).

---

## Round 0.3 · Minor Revisions

· Academic Editor

Minor Revisions

I think the objections raised have been successfully removed with one exception. In fact, while in the rebuttal letter it is stated:

8. l. 178-182: "Some highest … tRNAs": This sentence is still not clear. Please rephrase again.

> Answer: Accepted. The description in lines 178-182 were changed into the following: Some most used synonymous codons of PCGs were not corresponded to the tRNA anticodons of mitogenomes (Table S4). This results showed that the protein synthesis of mitogenomes need not only mitochondria encoded tRNAs, but also depend on nuclear encoded tRNAs.

in the revised manuscript the sentence does not correspond:

Some highest used codons by mitochondrial protein-coding genes among synonymous codons, such as UUU(F) and AUU(I), were not corresponded to the codons carried by mitogenomes encoded tRNAs (Table S4), which indicated that the mitochondrial protein synthesis of Caelifera need not only mitochondria encoded tRNAs, but a large extent depend on nuclear encoded tRNAs.

I do think the text of the rebuttal letter is preferable, although English usage could be improved.

I do hope this objection can be easily satisfied.

---

## Round 0.4 · accepted · Accept

· Academic Editor

Accept

I think the objection raised has been successfully removed.